

# The effects of meliponicultural use of *Tetragonula laeviceps* on other bee pollinators and pollination efficacy of lemon

Muhamad Aldi Nurdiansyah[1], Muhammad Yusuf Abduh[2], Aos Aos[2], Asep Hidayat[2] and Agus Dana Permana[2]

[1] Doctoral Program of Biology, School of Life Sciences and Technology, Institut Teknologi Bandung, Bandung, West Java, Indonesia
[2] School of Life Sciences and Technology, Institut Teknologi Bandung, Bandung, West Java, Indonesia

## ABSTRACT

The augmentation of pollination success in lemon (*Citrus limon* Eureka) flowers remains contingent on the involvement of bee pollinators. With wild bee pollinator populations declining in agroecosystems, meliponiculture has emerged as a potential option in Indonesia. This study aimed to investigate the effects of meliponicultural use of *Tetragonula laeviceps* on diversity, foraging behavior, and monthly population of bee pollinators, as well as lemon pollination efficacy with and without meliponiculture treatment during two periods. Using scan and focal sampling methods in first and second periods, the study found that the diversity of wild bee pollinators was six species (*Apis cerana*, *Lasioglossum albescens*, *Megachile laticeps*, *Xylocopa confusa*, *Xylocopa latipes*, and *Xylocopa caerulea*), and *T. laeviceps* when using meliponiculture. The relative abundance and daily foraging activity of wild bee pollinators were initially reduced in the first period (March–June) and then maintained in the second period (July–October). *T. laeviceps* foraged on the flowers, involving specific sequences for 72 s with highest visitation rate of 0.25 flowers/h from 10:00–13:00. Light intensity was observed to be the most influential factor for bee pollinator density. Pollination efficacy results showed that meliponiculture usage has greater benefit compared to meliponiculture absence across various parameters, including fruit sets, fruit weight, yield, and estimated productivity. The effects of meliponicultural use of *T. laeviceps* can enhance lemon pollination efficacy while preserving the diversity of wild insect pollinators. This suggests that meliponiculture stingless bees could be a beneficial practice in agroecosystems, especially in tropical regions where wild bee populations and diversity are declining.

## INTRODUCTION

Lemon (*Citrus limon* Eureka) is an agricultural fruit commodity grown in open farming with monoecious flower types that bloom all year. Lemon productivity in Indonesia is

Corresponding author
Muhammad Yusuf Abduh,
yusuf@sith.itb.ac.id

lower than the global average lemon productivity (*Badan Pusat Statistik Indonesia, 2022*; *Food & Agriculture Organization, 2023*). Wind pollination is sufficient to pollinate lemon flowers, while bee pollinators can ensure increased pollination success (*Aizen et al., 2019*; *Dymond et al., 2021*; *Vanlalhmangaiha et al., 2022*). However, numerous studies have reported that the population of wild bee pollinators is declining by 20–57% in various regions (*Potts et al., 2010*; *Koh et al., 2016*; *Rhodes, 2018*; *Panziera et al., 2022*; *MacInnis, Normandin & Ziter, 2023*). This issue has prompted several studies on applied bee cultivation in agroecosystems to enhance productivity (*Aslan & Yavuksuz, 2010*; *Nunes-Silva et al., 2013*; *Hall et al., 2020*; *Amon et al., 2023*). The application of meliponiculture has expanded in tropical regions with enhanced pollination efficacy of 10–70% (*da Silva et al., 2017*; *Azmi et al., 2019*; *Layek, Das & Karmakar, 2022*; *Reddy, Chauhan & Singh, 2022*; *Balaji et al., 2023*; *Wongsa, Duangphakdee & Rattanawannee, 2023*), particularly in Indonesia with 20–48% (*Putra, Permana & Kinasih, 2014*; *Alpionita, Atmowidi & Kahono, 2021*; *Atmowidi et al., 2022*; *Nurdiansyah, Abduh & Permana, 2023*).

In Indonesia, a total of 19 stingless bee species have been cultivated (*Buchori et al., 2022*), and six species have been utilized as pollinators in agricultural crops, *i.e.*, *Heterotrigona itama*, *Lepidotrigona terminata*, *Tetragonula iridipennis*, *Tetragonula biroi*, *Tetragonula clypearis*, and *Tetragonula laeviceps* (*Putra, Permana & Kinasih, 2014*; *Alpionita, Atmowidi & Kahono, 2021*; *Asmini, Atmowidi & Kahono, 2022*; *Djakaria, Atmowidi & Priawandiputra, 2022*; *Putra et al., 2022*; *Suhri et al., 2022*). *T. laeviceps* was the most commonly used species in meliponiculture in agricultural commodities such as true shallot, strawberry, okra, pummelo, and orange, and it became the main pollinator in orange orchards (*Nurdiansyah, Abduh & Permana, 2023*). Furthermore, *T. laeviceps* has been successfully cultivated across several islands including Sumatera, Java, Kalimantan, Sulawesi, Bali, and Maluku, with Java emerging as the predominant cultivation site and has been shown to enhance agricultural crop pollination success (*Asmini, Atmowidi & Kahono, 2022*; *Buchori et al., 2022*; *Djakaria, Atmowidi & Priawandiputra, 2022*). The effects of meliponiculture have been widely reported to improve quality and productivity of seeds and fruits, while to our knowledge is remains unreported whether it affects wild insect pollinators in open farming.

Maintaining the diversity of wild insect pollinators is critical for preserving ecological services, including the sustainability of natural habitats and population biodiversity (*Garibaldi et al., 2011*; *Tschoeke et al., 2015*). In agroecosystems, a rich diversity of wild insect pollinators contributed to enhanced pollination services (*Katumo et al., 2022*). However, the introduction of a new species into agroecosystems may lead to resource competition, potentially displacing existing wild bee pollinators from their role in pollination (*Nielsen et al., 2017*). A comprehensive study on meliponiculture in open farming is needed to address this issue.

This study aimed to investigate the effects of meliponicultural use of *T. laeviceps* on wild bee pollinators within a lemon orchard by conducting a comprehensive analysis over two harvest cycles, contrasting conditions with and without meliponiculture. The study commenced with an assessment of the diversity and relative abundance of wild bee
pollinators. Subsequently, it delved into exploring pollinator foraging behaviors, including daily activities, time spent, pollen load, and the influence of environmental factors on visitation rates. Additionally, the observation of pollination sequences focused solely on *T. laeviceps* when pollinating lemon flowers. Furthermore, the monthly population of bee pollinators was observed to elucidate the dynamic patterns of pollinator abundance and distribution. Moreover, this study also evaluated the pollination efficacy of *T. laeviceps* on fruit set, fruit weight, and lemon productivity. The findings of this study provide valuable insights into the application of meliponiculture to support sustainable agriculture.

# MATERIALS AND METHODS

## Study site

This study was carried out in Cibodas, West Bandung Regency, West Java, Indonesia, at coordinates latitude 6°49′20″S and longitude 107°40′35″E, with an altitude of 1,219 meters above sea level. The study covered a total land area of $60 \times 60$ m$^2$ and included 200 plants of lemon (*Citrus limon* Eureka) aged 3 years, with an average height of 2 m and a canopy width of 2 m. The eastern, southern, and western parts of the lemon orchard are surrounded by horticultural farming including eggplants, tomatoes, and cabbage, while the northern part is bordered by teak forests. The colony carrying capacity in lemon has been calculated and the results require four colonies of stingless bee *Tetragonula laeviceps*, with approximately 400 to 600 adult worker bees for each colony (*Bareke et al., 2020*). The colonies were obtained from cultivators in Banjaran, West Java, Indonesia and acclimatized for 1 week before the observations.

The study was investigated in two periods, with the first period from March to June 2023 and the second period from July to October 2023, with four cultivated colonies of *T. laeviceps* in Modular Tetragonula Hives (MOTIVEs) with size $20 \times 20 \times 15$ cm$^3$ (*Abduh et al., 2020*). The MOTIVEs were placed in the middle of a lemon orchard in late March and left until October 2023. The study employed four plots placed in accordance with the compass, each measuring $15 \times 15$ m$^2$, with 16 lemon plants per plot. Each plant received 5 kg of organic fertilizer made from chicken manure in March and July 2023. Observations were conducted from 07:00 to 15:00, employing 15 min intervals for each plot. Observations for plots without meliponiculture were conducted weekly throughout March 2023, whereas observations for plots with meliponiculture were undertaken weekly from April to October 2023. Microclimate conditions at the study site were measured using the Data Logger HOBO U10-003.

## Diversity, foraging behavior, and population of bee pollinators

Wild insect pollinators were collected by sweep net (mesh size $0.9 \times 0.3$ mm$^2$) using a dried preservation technique and subsequently pinned. Furthermore, lemon flowers and bee pollinators carrying pollen were captured and inserted into a 25 mL colonial tube containing 15 mL of 70% alcohol. These specimens were sent to the Laboratory of Entomology, School of Life Sciences and Technology, Institut Teknologi Bandung, Indonesia for taxonomic identification (*Gibb & Oseto, 2019*; *Mason et al., 2022*) and number of lemon pollen and pollen load of bee pollinators analysis. After determining the

taxonomic identification of pollinators, the diversity of pollinators was analyzed using the percentage of relative abundance and Eq. (1).

$$Relative\ abundance\ (\%) = \frac{Total\ number\ of\ each\ pollinator\ species}{Total\ number\ of\ pollinators} \times 100\%. \qquad (1)$$

Foraging behavior of bee pollinators was observed using a scan sampling method across four plots, with each plot observed for 15 min to determine the number of bees visiting blooming lemon flowers, while the time spent by bee pollinators per flower was recorded using a focal sampling method, which involves directly observing each pollinator from the present moment it visits the flower until it leaves (*Putra, Permana & Kinasih, 2014*; *Nurdiansyah, Abduh & Permana, 2023*). Lemon flowers were wrapped in an insect net (mesh size 36 mm) before they bloomed in the second period. When they bloomed fully, nectar was collected using micro hematocrit capillaries (length 75 mm, diameter 1.55 mm) each hour from 07:00 to 16:00. If nectar was not collected from the flowers, they were wrapped again in an insect net to prevent insects from accessing the nectar. Volume was measured with a micropipette, and the concentration was determined using a hand refractometer.

For pollen analysis, flowers or bees were centrifuged for 5 min at 3,500 rpm, and then the flowers or bees were removed. Subsequently, another centrifugation was performed for 3 min at 2,000 rpm, and the supernatant was removed. A solution of acetolysis (0.9 mL acetic anhydride + 0.1 mL sulfuric acid) was added, and the samples were heated in a water bath at 80 °C for 5 min. Afterward, 1 mL of distilled water was added. The number of pollen grains was quantified using 0.1 mL of samples at hemocytometer in four quadrants under a light microscope (eyepiece lens 10× and objective lens 10×/0.24) (Nikon Instruments, Tokyo, Japan). The number of pollens grains from flowers and bees was calculated using the total volume of solution multiplied by the number of pollen grains counted and divided into a volume of four quadrants (*Alpionita, Atmowidi & Kahono, 2021*).

The foraging behaviors of bee pollinators were assessed based on visitation rates, which were categorized into three intervals: morning (7:00 to 10:00), noon (10:00 to 13:00), and afternoon (13:00 to 16:00). The visitation rates of bee pollinators was calculated as the number of bee pollinators visiting flowers divided by the number of flowers available per observation (*Gallagher & Campbell, 2020*). The total number of bees visiting blooming flowers of each species and the total number of blooming flowers per month were utilized to analyze the monthly population of bee pollinators.

## Pollination efficacy and productivity estimation of lemon

Pollination efficacy was investigated in two periods, with the first period from flower until harvest observed from March to June 2023, and the second period observed from July to October 2023. Pollination efficacy was compared between without and with *T. laviceps* based on various parameters, including number of fruit sets, pollination success, fruit weight, and yield per plant in the first period, as well as with meliponiculture in the first and second period. The comparison involves 15 flowers per plant from 64 plants that were

randomly tagged, and after days of blooming stages, the number of fruit sets was calculated. Percentage of fruit set was determined by calculating as the number of fruit sets divided by the number of flowers, multiplied by 100% (*Nurdiansyah, Abduh & Permana, 2023*). When harvesting, three lemons were selected at random from each plant and weighed using a digital analytical balance. The yield per lemon plant was determined by multiplying the average fruit weight by the number of harvested fruits. Lemon productivity was also estimated per hectare given there are 900 plants using a $3 \times 3$ m$^2$ spacing. The estimated lemon productivity was determined by multiplying the average fruit weight by the number of fruits harvested, the number of plants, and accounting for three harvest cycles within a single year (*Nurdiansyah, Abduh & Permana, 2023*).

## Statistical analysis

All data were analyzed for normality and homogeneity of variance, and no data transformations were applied. Effects of *T. laeviceps* were analyzed using the two-sample t-test ($p < 0.05$) for the parameter's relative abundance, daily foraging activity, fruit sets, fruit weight, and yield per plant. Analysis of variance (ANOVA) followed by Tukey's multiple comparison test ($p < 0.05$) were performed to compare the pollen load, time spent, and visitation rates of bee pollinators. Pearson's correlation coefficient was employed to assess the significance of the correlation between the number of pollinators visiting lemon flower with environmental factors. Additionally, monthly correlation calculations were conducted to examine the relationship between the monthly population of bee pollinators and the number of flowers. To identify the primary components influencing the foraging behavior of bee pollinators on lemon flowers, a principal component analysis (PCA) was performed. The statistical analyses were performed using the R program version 4.3.2. (*R Core Development Team, 2023*).

# RESULTS

## Diversity of insects visiting lemon flowers

Without meliponiculture, wild insects visiting lemon flowers were six species of bees, *i.e.*, *Apis cerana* (Hymenoptera: Apidae), *Lasioglossum albescens* (Hymenoptera: Halictidae), *Megachile laticeps* (Hymenoptera: Megachilidae), *Xylocopa confusa* (Hymenoptera: Apidae), *Xylocopa latipes* (Hymenoptera: Apidae), and *Xylocopa caerulea* (Hymenoptera: Apidae), along with four species of non-bees, including *Dolichoderus thoracicus* (Hymenoptera: Formicidae), *Papilio demoleus* (Lepidoptera: Papilionidae), *Delias belisama* (Lepidoptera: Pieridae), and *Hypolimnas misippus* (Lepidoptera: Nymphalidae). However, the diversity of wild insect visitors with meliponiculture was maintained, and *Tetragonula laeviceps* (Hymenoptera: Apidae) emerged as a new pollinator for lemon flowers with the highest relative abundance ($t_{(3)} = 60.00$, $p = 1.02\text{E}^{-5}$). The relative abundance of wild bees visiting lemon flowers decreased significantly in the first period, including *A. cerana* ($t_{(6)} = 31.84$; $p = 6.38\text{E}^{-8}$), *L. albescens* ($t_{(6)} = 28.54$; $p = 1.23\text{E}^{-7}$), *M. laticeps* ($t_{(6)} = 8.69$; $p = 1.28\text{E}^{-4}$), *X. confusa* ($t_{(6)} = 31.72$; $p = 6.52\text{E}^{-8}$), *X. confusa* ($t_{(6)} = 31.72$; $p = 6.52\text{E}^{-8}$), *X. caerulea* ($t_{(6)} = 18.04$; $p = 1.87\text{E}^{-6}$), while the non-bees remained constant (Table 1). There was no change in the relative abundance of wild insect

**Table 1 Percentage relative abundance (%) of insect visitors at first (I) and second (II) periods on lemon flowers.**

| Insect visitors | Without meliponiculture | With meliponiculture | |
|---|---|---|---|
| | I ($n$ = 4) | I ($n$ = 4) | II ($n$ = 4) |
| Bees | | | |
| *T. laeviceps* | 0b | 41.66 ± 0.38a | 41.75 ± 0.32a |
| *A. cerana* | 56.68 ± 0.62a | 30.85 ± 0.36b | 30.70 ± 0.31b |
| *L. albescens* | 23.11 ± 0.66a | 13.38 ± 0.27b | 13.41 ± 0.22b |
| *M. laticeps* | 8.02 ± 0.45a | 6.64 ± 0.17b | 6.64 ± 0.14b |
| *X. confusa* | 4.63 ± 0.38a | 2.53 ± 0.14b | 2.48 ± 0.13b |
| *X. latipes* | 3.17 ± 0.21a | 1.88 ± 0.12b | 1.87 ± 0.13b |
| *X. caerulea* | 3.19 ± 0.22a | 1.86 ± 0.12b | 1.85 ± 0.11b |
| Non-bees | | | |
| *D. thoracicus* | 0.67 ± 0.05a | 0.66 ± 0.04a | 0.68 ± 0.04a |
| *P. demoleus* | 0.21 ± 0.03a | 0.20 ± 0.04a | 0.21 ± 0.04a |
| *D. belisama* | 0.18 ± 0.02a | 0.18 ± 0.02a | 0.17 ± 0.02a |
| *H. misippus* | 0.15 ± 0.01a | 0.15 ± 0.02a | 0.15 ± 0.02a |

Note:
Data are means ± SD followed different letters within a row indicate significant differences ($p < 0.05$) by two-sample t-test. $n$ is the number of observations.

visitors between the first period with meliponiculture and second period with meliponiculture ($p > 0.05$).

## Foraging behavior of bee pollinators

The daily foraging activity of bee pollinators visiting lemon flowers starts from 7:00 to 16:00 (Fig. 1). The highest number of bee pollinators visiting lemon flowers occurred at 11:00 in both the first and second periods. However, the highest number of bee pollinators visiting lemon flowers in the first period significantly decreased between without and with meliponiculture, including *A. cerana* ($t_{(6)} = 85.90$; $p = 1.56\text{E}^{-10}$), *L. albescens* ($t_{(6)} = 48.18$; $p = 5.36\text{E}^{-9}$), *M. laticeps* ($t_{(6)} = 51.30$; $p = 3.68\text{E}^{-9}$), *X. confusa* ($t_{(6)} = 47.82$; $p = 6.60\text{E}^{-9}$), *X. latipes* ($t_{(6)} = 18.59$; $p = 1.56\text{E}^{-6}$), and *X. caerulea* ($t_{(6)} = 18.64$, $p = 1.54\text{E}^{-6}$). There were no significant differences in daily foraging activity between with meliponiculture in the first and with meliponiculture in the second period ($p > 0.05$).

Each visit to lemon flowers by *T. laeviceps* follows a distinct pollination sequence (Fig. 2). It starts with approaching (Fig. 2A) the flower at a position parallel to 45 degrees from its original position, followed by perching on a petal (Fig. 2B) or an anther (Fig. 2C). Subsequently, it enters the nectary flower to collect nectar (Fig. 2D), then climbs to the anther to collect pollen (Fig. 2E). When solely interested in pollen, *T. laeviceps* directly perching goes to the anther. Upon completing its activities, it leaves the lemon flower from the anther (Fig. 2F) without buzzing and moves to search for the nearest flower to resume collecting nectar and pollen (Fig. 2G). *T. laeviceps* exhibited the longest time visiting lemon flower ($F_{(6, 105)} = 12.22$; $p = 0.000$), while *M. laticeps* exhibited the fastest time ($F_{(6, 105)} = 26.01$; $p = 0.000$) (Table 2).

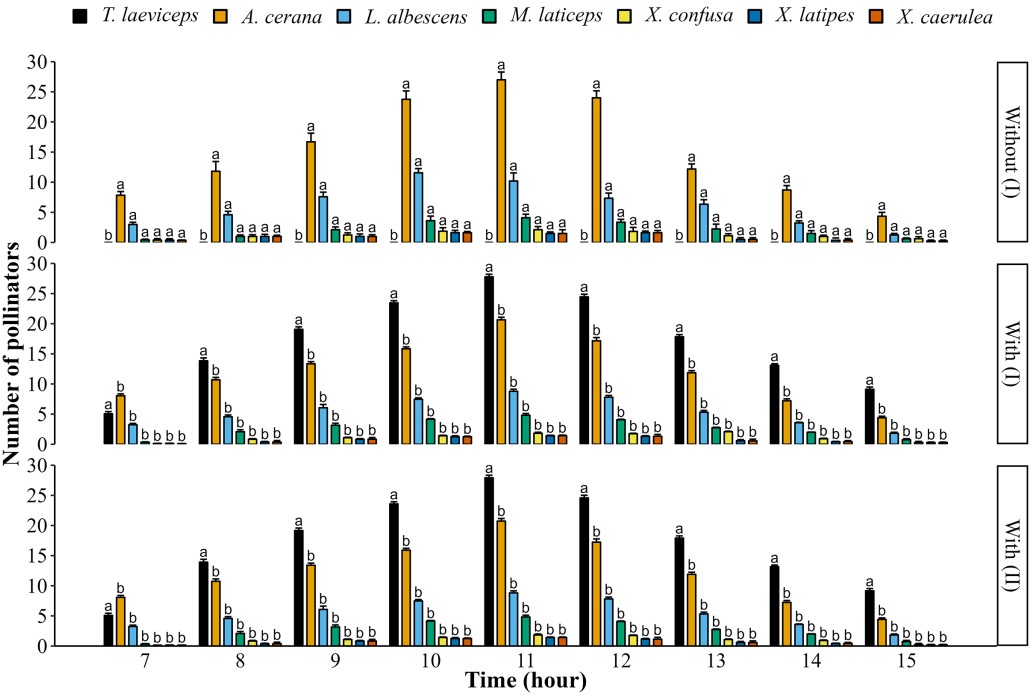

**Figure 1 Daily foraging activity of bee pollinators in the first (I) and second (II) period.** Data are means ± SD followed by different letters within a row which indicate significant differences ($p < 0.05$) using the two-sample t-test ($n = 4$).

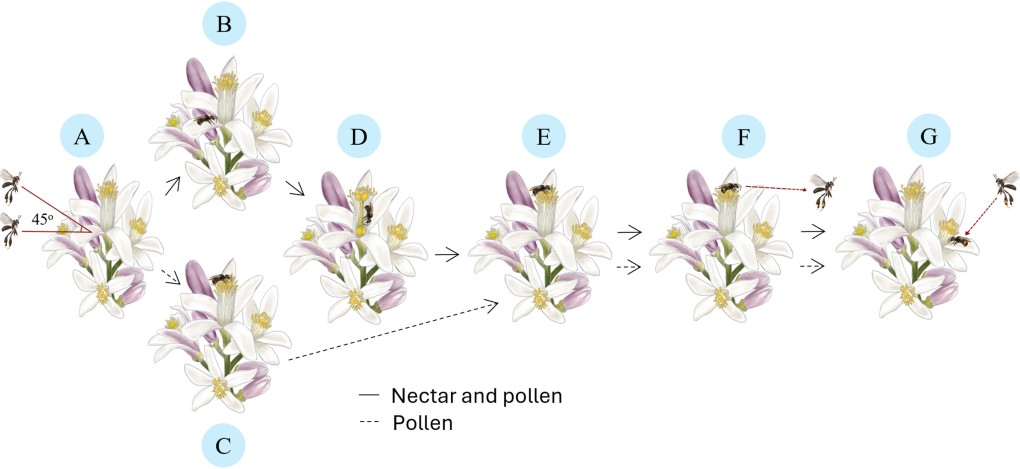

**Figure 2 Pollination sequences of *Tetragonula laeviceps* on lemon flowers.** (A) Approaching: orienting in a position parallel to the 45 degrees flower position. (B) Perching: on the petal for nectar and pollen. (C) Perching: on the anther for pollen. (D) Collecting: entering the nectary flower for nectar. (E) Collecting: climbing into the anther for pollen. (F) Leaving: from the anther. (G) Moving: to the nearest flower to resume collecting nectar and pollen. Image credit: The authors hired a freelance illustrator name: Arsa Admahira.

Lemon flowers produce 12,539 ± 376 pollen grains per flower. Each bee pollinator carries pollen on its body after visiting a lemon flower and the pollen load of each bee pollinator differs ($p < 0.05$). Based on pollen load on their bodies, bee pollinators seem to

**Table 2 Pollen load (pollen grains) and time spent (s/flower) of bee pollinators.**

| Bee pollinators | Pollen load (pollen grains) | | Time spent (s/flower) |
| --- | --- | --- | --- |
| | Lemon ($n = 16$) | Miscellaneous ($n = 16$) | Lemon ($n = 16$) |
| *T. laeviceps* | 6,124.23 ± 115.45c | 127.35 ± 2.67f | 71.32 ± 5.64a |
| *A. cerana* | 84,875.76 ± 324.39a | 258.59 ± 4.33d | 59.31 ± 4.23c |
| *L. albescens* | 6,213.43 ± 96.62c | 285.84 ± 6.67b | 62.53 ± 5.37b |
| *M. laticeps* | 13,125.28 ± 121.65b | 504.21 ± 5.33a | 45.76 ± 3.89e |
| *X. confusa* | 873.46 ± 23.21d | 278.93 ± 2.13b | 52.74 ± 3.34d |
| *X. latipes* | 867.94 ± 22.98d | 249.47 ± 3.67e | 54.21 ± 2.98d |
| *X. caerulea* | 883.71 ± 32.64d | 253.56 ± 5.56e | 53.75 ± 3.86d |

Note:
Data are means ± SD followed different letters within a column indicate significant differences ($p < 0.05$) by ANOVA with Tukey's HSD test. $n$ is the number of bee pollinators.

visit not only lemon flowers but also other flowers in the lemon orchard, such as tomatoes and eggplants. The honey bee *A. cerana* has the highest pollen load in lemon flowers, with 84,875 pollen grains ($F_{(6, 105)} = 71,728.25$; $p = 0.000$), while the stingless bee *T. laeviceps* carries only 6,124 pollen grains ($F_{(6, 105)} = 5,239.94$; $p = 0.000$). Furthermore, *M. laticeps* ($F_{(6, 105)} = 12,372.51$; $p = 0.000$) carries the highest number of pollen grains from various plant flowers. The volume of nectar on lemon flowers continued to increase from 7:00 to 11:00 and decreased until 16:00 when the last bee pollinators visited lemon flowers (Fig. 3A). However, the nectar concentration continued to increase from 7:00 to 16:00. The highest pollinator visitation rates occurred at noon (10:00 to 13:00), with *T. laeviceps* being the most significant pollinator visiting lemon flowers ($F_{(12, 483)} = 7.42$; $p = 0.000$). However, it was observed that the lowest pollinator visitation rate was for *X. caerulea* ($F_{(12, 483)} = 0.01$; $p = 0.000$) during the afternoon (13:00 to 16:00).

The number of bee pollinators visiting lemon flowers was then analyzed in correlation with environmental factors such as microclimate conditions and nectar contents. Microclimate conditions including temperature, light intensity, and relative humidity during observations from March to October 2023, ranging from 20.27–24.29 °C, 589.3–5,442.5 lux, and 71.95–87.43%, respectively. The temperature followed the same pattern as the number of pollinators and also showed a very high positive correlation ($r = 0.83$; $p = 2.22\text{E}^{-16}$), as with performed light intensity ($r = 0.93$; $p = 2.22\text{E}^{-16}$) (Fig. 4). Whereas relative humidity follows the opposite pattern and showed a high negative correlation ($r = -0.68$; $p = 3.26\text{E}^{-11}$). In addition, the number of bee pollinators visiting lemon flowers also correlates with nectar content, with a high positive correlation with volume ($r = 0.65$; $p = 6.20\text{E}^{-10}$), and a low positive correlation with concentration ($r = 0.24$; $p = 0.04$). Furthermore, a principal component analysis (PCA) was performed to explore the relationship between microclimate conditions, nectar content, and the number of pollinators visiting lemon flowers (Fig. 5). The results indicated that light intensity (Dim1 = 2.29; Dim2 = −0.36) is the most significant component that influenced on the number of pollinators (Dim1 = 2.33; Dim2 = −0.02), followed by temperature

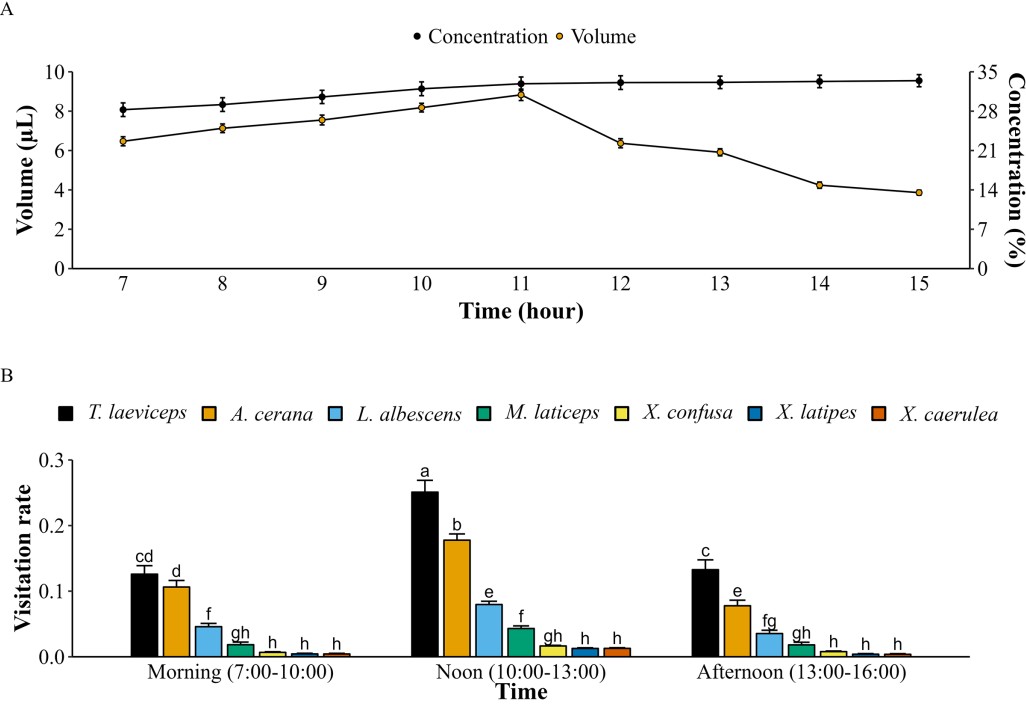

**Figure 3 Nectar content of lemon flowers and visitation rate of bee pollinators with meliponiculture.** (A) Volume and concentration of nectar ± SD ($n = 16$), (B) pollinators visitation rates on lemon flowers ± SD followed different letters within a bar indicate significant differences ($p < 0.05$) using ANOVA with Tukey's HSD test ($n = 16$).

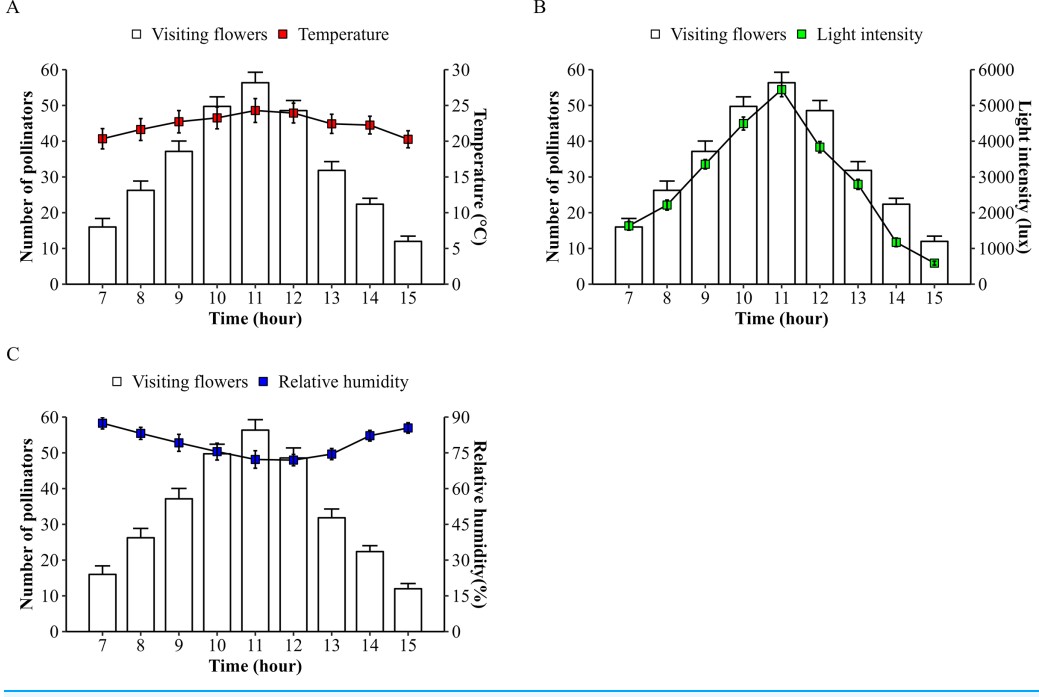

**Figure 4 Microclimate conditions during foraging behaviors of insect pollinators visiting lemon flowers.** (A) Temperature ± SD, (B) light intensity ± SD, and (C) relative humidity ± SD.

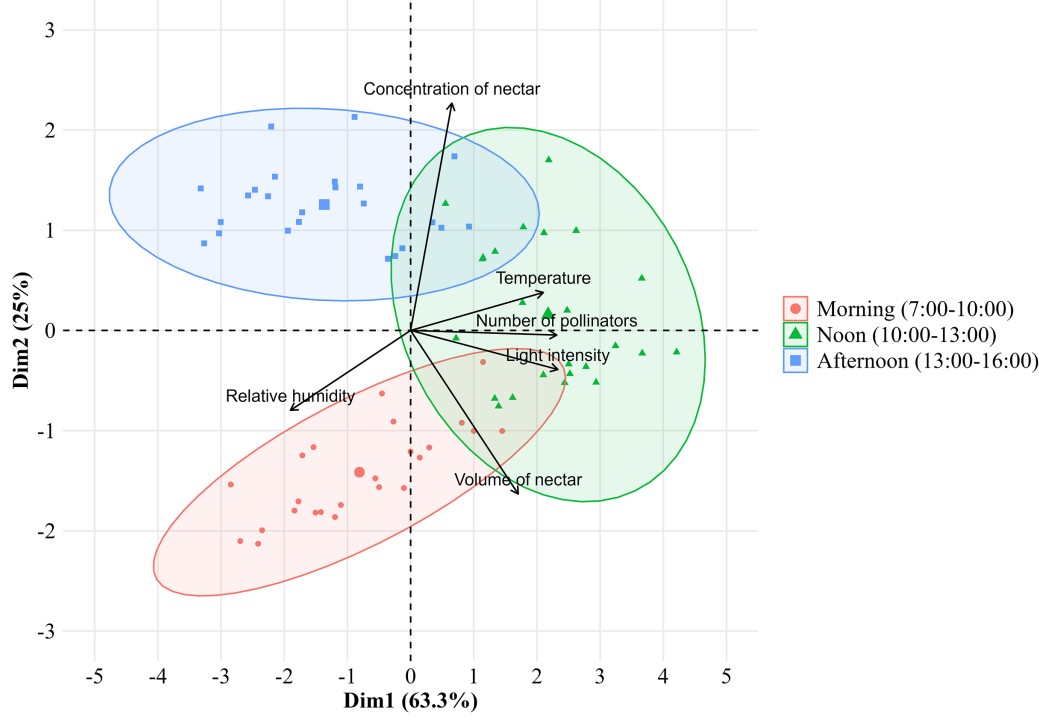

**Figure 5 Principal component analysis between the environmental factors with number of pollinators visiting lemon flowers.**

(Dim1 = 2.08; Dim2 = 0.34), relative humidity (Dim1 = −1.89; Dim2 = −0.87), volume (Dim1 = 1.71; Dim2 = −1.62) and concentration of nectar (Dim1 = 0.74; Dim2 = 2.26).

## Population of bee pollinators

The monthly population of bee pollinators visiting lemon flowers shows fluctuations (Fig. 6). The population of bee pollinators and lemon flowers increased from March to April in the first period, but then decreased until June. In the second period, a similar pattern was observed. This indicates that the lemon production cycle lasts for 4 months, with phenological stages lasting 115–120 days in the first period and 121–125 days in the second period from full bloom to harvest from 20 flowers observed. The total number of bee pollinators exhibited a very high positive correlation ($r = 0.98$; $p = 2.25\text{E}^{-5}$) with the total number of blooming lemon flowers.

## Pollination efficacy and productivity estimation of lemon

The pollination efficacy on 64 lemon plants was evaluated using 15 flowers per plant (Table 3). The results of the first period showed a significant difference in the fruit set ($t_{(126)} = 26.47$; $p = 1.29\text{E}^{-52}$), fruit weight ($t_{(126)} = 118.49$; $p = 4.44\text{E}^{-131}$), and yield per plant ($t_{(126)} = 108.63$; $p = 2.27\text{E}^{-126}$) between without and with *T. laviceps*. There were no significant differences in the parameters of fruit sets, fruit weight, and yield per plant in both sampling periods with *T. laviceps* ($p > 0.05$). Estimated lemon productivity without meliponiculture is $11.62 \pm 0.24$ tons per hectare per year, while with meliponiculture is

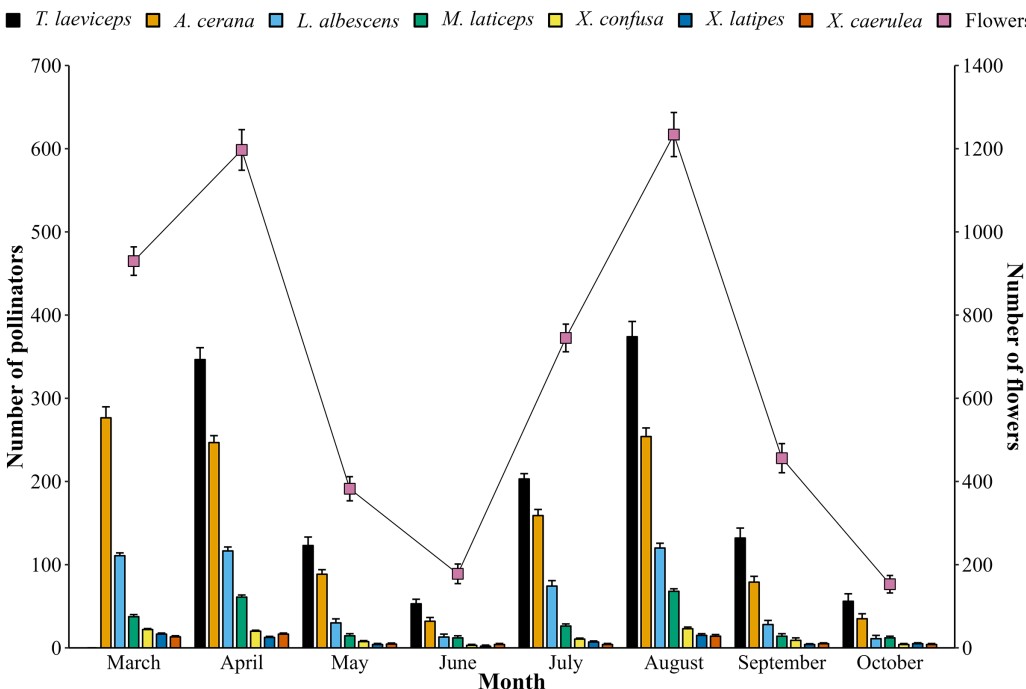

**Figure 6 Monthly population of bee pollinators (means ± SD) and lemon flowers (means ± SD).**

**Table 3 Effects of meliponiculture *Tetragonula laeviceps* pollination on fruit set (%), fruit weight (g), and yield (g/plant) of lemon.**

| Parameters | Without meliponiculture | With meliponiculture | |
| --- | --- | --- | --- |
| | I (n = 64) | I (n = 64) | II (n = 64) |
| Number of flowers | 15a | 15a | 15a |
| Fruit set (%) | 61.67 ± 2.91b | 76.25 ± 3.331a | 76.98 ± 3.34a |
| Fruit weight (g) | 140.56 ± 2.86b | 147.53 ± 2.69a | 147.78 ± 2.57a |
| Yield (g/plant) | 844.20 ± 11.17b | 1,183.40 ± 22.34a | 1,183.00 ± 22.45a |

**Note:**
Data are means ± SD followed different letters within a row indicate significant differences ($p < 0.05$) by the two-sample t-test. *n* is the number of lemon plants.

15.19 ± 0.12 tons per hectare per year in the first period and 14.92 ± 0.16 tons per hectare per year in the second period.

## DISCUSSION

This study showed that meliponiculture of *Tetragonula laeviceps* in lemon orchards does not affect the diversity of wild insects visiting flowers, including wild bees and non-bees in both periods, as with meliponiculture in orange orchards (*Nurdiansyah, Abduh & Permana, 2023*). However, the relative abundance of wild bees decreased (42%), while the relative abundance of wild non-bees was maintained in the first period. This demonstrates the existence of competition between wild bees and *T. laeviceps*, leading to *T. laeviceps*

being the most abundant pollinator of lemon flowers. Similar patterns have been observed with managed honeybees, which can reduce the density of wild bumble bees in homogeneous plant landscapes and raspberry farming (*Herbertsson et al., 2016*; *Nielsen et al., 2017*). This suggests that stingless bees *T. laeviceps* show high fidelity to lemon flowers followed by wild honeybees *A. cerana*. The consistent relative abundance between the first and second periods suggests the structure of an insect pollinator community is stable, and the meliponiculture *T. laeviceps* has no potential to disrupt the community structure in the short-term. A study was required to evaluate the potential long-term disruptiveness of meliponicultural use of *T. laeviceps*, because honeybees disrupted the structure of plant-pollinator interactions (*Valido, Rodríguez-Rodríguez & Jordano, 2019*).

The daily foraging activity of wild bee pollinators visiting lemon flowers was reduced in the first period with meliponiculture *T. laeviceps*. The decline in wild bee pollinators can be attributed to flower constancy, which causes bee species to avoid flowers previously visited by colony bees (*Grüter & Ratnieks, 2011*; *Nielsen et al., 2017*). This is supported by the behavior of colony bees such as stingless bees, which leave a pheromone trail on flowers to indicate that they have been visited (*Jarau et al., 2010*, *2011*; *Grüter, 2020*). The daily foraging activity of *T. laeviceps* on lemon flowers peaked at 11:00 each day in meliponiculture periods. This finding is supported by the highest number of *T. laeviceps* entering and exiting the hive (*Abduh et al., 2023*). However, the highest foraging activity in *Tetragonula pagdeni* was at 10:00 in greenhouse conditions (*Wongsa, Duangphakdee & Rattanawannee, 2023*). Furthermore, several studies reported that at 11:00, *T. laeviceps* was the visiting mostly strawberry, mango, and orange flowers (*Atmowidi et al., 2022*; *Chuttong et al., 2022*; *Nurdiansyah, Abduh & Permana, 2023*).

The pollination sequences of *T. laeviceps* commence with the worker bees positioned parallel to or 45 degrees above the flower. This positioning is thought to be related to the flight method of *T. laeviceps*, which avoids flying over its resource's plants. Following that, worker bees approach lemon flowers directly, guided most likely by scouting bees that had marked the locations of flowers containing nectar and pollen (*Grüter, 2020*). When collecting nectar, *T. laeviceps* lands on the petals and then enters the nectary of the flower, while when collecting pollen, *T. laeviceps* lands directly on the anthers. *T. laeviceps* spends more time collecting resources in lemon flowers than honey bees, and the same occurs with other flower plants (*Putra, Permana & Kinasih, 2014*; *Alpionita, Atmowidi & Kahono, 2021*; *Nurdiansyah, Abduh & Permana, 2023*). This is due to the small size (±0.5 cm) of *T. laeviceps* and its opportunistic approach to carrying as much as possible on its body, which can contain approximately 6,200 lemon pollen grains. This pollen load in lemon flowers was lower compared to other plants, such as strawberries with 8,600 pollen grains, and melons with 26,200 pollen grains (*Alpionita, Atmowidi & Kahono, 2021*; *Bahlis, Atmowidi & Priawandiputra, 2021*). Subsequently, *T. laeviceps* departs from the flower *via* the anthers, facilitating pollen transfer to the stigma and enhancing pollination success, thus reaping the benefits of the plant-pollinator interaction. *T. laeviceps* pollination sequences are similar to honey bees but differ slightly from bumble bees, which leave plant flowers buzzing near the anthers. This behavioral variation may affect pollination efficacy,
as bumble bees' buzzing behavior is thought to improve pollination effectiveness (*Heard, 1994*; *Nunes-Silva et al., 2013*; *Cooley & Vallejo-Marín, 2021*).

The highest pollinator visitation rate for lemon flowers occurred at noon (10:00 to 13:00), with *T. laeviceps* being the most frequent visitor at 0.25 flowers/h, followed by *A. cerana* visiting at 0.18 flowers/h, which was consistent with a previous study on meliponiculture of *T. laeviceps* in orange orchards (*Nurdiansyah, Abduh & Permana, 2023*). This can be attributed to the first full bloom of lemon flowers around 10:00, followed by an increase in the volume of nectar secretion. In the afternoon (13:00 to 16:00), the volume of nectar decreased, followed by a decline in the pollinator visitation rate. However, the nectar concentration increased during this period. The nectar secretion pattern of lemon flowers is similar to that of *Croton macrostachyus* flowers (*Bareke et al., 2020*). The flowers were rarely visited by bee pollinators on the second day, indicating that the rewards offered by lemon flowers had decreased because the volume and sugar content of nectar decreased over time (*Chauhan, Chauhan & Galetto, 2017*). This is supported by the Pearson's correlation analysis, indicating that the nectar contents including volume ($r = 0.65$) and concentration ($r = 0.24$) had a positive correlation with the number of bee pollinators visiting lemon flowers.

Following that, the microclimate conditions were investigated, and it was demonstrated that temperature ($r = 0.83$) and light intensity ($r = 0.93$) had a positive correlation with the number of pollinators, while the relative humidity ($r = -0.68$) had a negative correlation. These findings are consistent with previous studies indicating that temperature, light intensity, and relative humidity are factors influencing the foraging behavior of bee pollinators (*Polatto, Chaud-Netto & Alves-Junior, 2014*), honeybees (*Taha, Al-Abdulsalam & Al-Kahtani, 2016*), and *T. laeviceps* (*Abduh et al., 2023*; *Nurdiansyah, Abduh & Permana, 2023*). The principal component analysis (PCA) revealed that light intensity is the most influential environmental factor affecting the number of pollinators visiting lemon flowers. This finding contradicts previous studies, which emphasized that temperature was the predominant factor in visits to flower during pollinator activity (*Taha, Al-Abdulsalam & Al-Kahtani, 2016*; *Gallagher & Campbell, 2020*; *Layek, Kundu & Karmakar, 2020*), although temperature significantly influences the activities of stingless bee *Plebeia* aff. *flavocincta* outside the hive (*Barbosa et al., 2020*). However, observations on stingless bee *Tetragonula pagdeni* suggest that foraging activity in collecting pollen from tomatoes plants increases under stable temperature conditions (*Wongsa, Duangphakdee & Rattanawannee, 2023*). These findings suggest that when temperature conditions are relatively stable, other microclimate conditions such as light intensity can play an important role in augmenting bee pollinator activity, especially regarding stingless bees during resource collection. The foraging behavior of bee pollinators is shaped by the intricate interaction of environmental factors in lemon orchards.

The monthly population of bee pollinators in lemon orchards showed a positive correlation ($r = 0.98$) with the number of blooming flowers. Bee pollinators visit a single lemon flower 1.41–1.95 times per day, indicating that lemon flowers provide a resource-rich environment for pollinators, including pollen and nectar. Comparing the pollen load of *T. laeviceps* (6,124 pollen grains) to the amount of pollen of lemon flowers

(12,539 pollen grains) suggests that each lemon flower could potentially be visited by at least two stingless bees. However, the nectar load of *T. laeviceps* and other stingless bee species remain unknown. In contrast, honey bees are reported to carry 22 µL of sugar syrup (50%) in 1 day (*Huang, 2018*), and the nectar from a lemon flower (58.54 µL) can potentially sustain at least two honey bees. It was observed that *T. laeviceps* visited lemon flowers 3.39 times per flower in one day. This shows that foraging behavior of *T. laeviceps* requires more flower visits to fill its body with pollen compared to the available amount of pollen, potentially influencing the pollination success of lemon flowers.

Pollination efficacy of meliponicultural use of *T. laeviceps* produces more fruit sets (15%) compared without meliponiculture, and these findings are consistent with previous studies in open farming (*Layek et al., 2021*; *Chuttong et al., 2022*; *Chauhan & Singh, 2022*; *Balaji et al., 2023*; *Nurdiansyah, Abduh & Permana, 2023*), and closed farming (*Azmi et al., 2019*; *Moura-Moraes et al., 2021*; *Layek, Das & Karmakar, 2022*; *Reddy, Chauhan & Singh, 2022*). According to *Gallagher & Campbell (2020)*, there is a positive correlation between higher pollinator diversity and larger pollinator populations, leading to increased pollinator visitation rates in agricultural landscapes, with potential implications for enhancing pollination success.

Meliponiculture using *T. laeviceps* contributes to producing more fruits which are larger and heavier. However, it is essential to note that specific impact on seed formation in lemon was not quantified in the present study. Numerous studies have suggested that variations in fruit weight attributed to different pollination methods are often associated with the number of seeds formed during pollination process, influenced by the pollinator and the frequency of visits (*Gallagher & Campbell, 2020*; *Azmi et al., 2022*; *Wongsa, Duangphakdee & Rattanawannee, 2023*). A notable 28–34% of fruits drop from the initial fruit set to the harvested stage in both treatments. Incorporating meliponiculture results in notable distinctions during the initial phases of fruit set and an augmented fruit weight, leading to an increased yield of lemon plants. Consequently, a higher fruit set corresponds to a greater overall yield. Although, the computed estimate lemon productivity in the present study is below the world average productivity in 2021 of 15.56 tons per hectare (*Food & Agriculture Organization, 2023*). Nonetheless, the estimated lemon productivity with meliponiculture closely aligns with world productivity, reaching 15 tons per hectare per year, representing a 23% increase compared to cultivation without meliponiculture. Meliponiculture emerges as an excellent option for enhancing the fruit set of lemon with consequences on lemon productivity in tropical regions.

## CONCLUSIONS

The meliponicultural use of *Tetragonula laeviceps* did not affect pollinator diversity in both periods, while the relative abundance and daily foraging activity of wild bee pollinators were reduced in the first period and then maintained in the second period. Pollination sequences of *T. laeviceps* involve approaching the lemon flower from a parallel position or 45° angle above the flower position by perching on the petal or anther, with a time spent of 71.32 ± 5.64 s in collecting nectar and pollen, and then consistently leaving the lemon

flower specifically from the anther. *T. laeviceps* exhibited the highest pollinator visitation rate of 0.25 flowers/h in the noon period (10:00 to 13:00), while *X. caerulea* had the lowest rate with 0.01 flowers/h. The number of bee pollinators visiting lemon flowers are influenced by environmental factors, with light intensity being the most influencing factor. Pollination efficacy with meliponiculture of *T. laeviceps* produces 15% more fruit sets and 23% more estimated productivity than without meliponiculture. This study suggests that meliponiculture stingless bees could be beneficial as pollinators in agricultural farming while maintaining pollinator diversity, which is critical for sustainable agriculture and enhanced pollination efficacy and productivity.

## ACKNOWLEDGEMENTS
We would like to thank all members of the Laboratory of Entomology, School of Life Sciences and Technology, Institut Teknologi Bandung.

### Funding
This work was supported by the Research, Community Service, and Innovation (PPMI), School of Life Sciences and Technology, Institut Teknologi Bandung (ITB) No: 27A/IT1. C11/SK-TA/2023. The funders had no role in study design, data collection and analysis, decision to publish, or preparation of the manuscript.

### Grant Disclosures
The following grant information was disclosed by the authors:
Research, Community Service, and Innovation (PPMI), School of Life Sciences and Technology, Institut Teknologi Bandung (ITB): 27A/IT1.C11/SK-TA/2023.

### Competing Interests
The authors declare that they have no competing interests.

### Author Contributions
- Muhamad Aldi Nurdiansyah conceived and designed the experiments, performed the experiments, analyzed the data, prepared figures and/or tables, authored or reviewed drafts of the article, and approved the final draft.
- Muhammad Yusuf Abduh conceived and designed the experiments, analyzed the data, authored or reviewed drafts of the article, and approved the final draft.
- Aos Aos analyzed the data, authored or reviewed drafts of the article, and approved the final draft.
- Asep Hidayat analyzed the data, authored or reviewed drafts of the article, and approved the final draft.
- Agus Dana Permana conceived and designed the experiments, analyzed the data, authored or reviewed drafts of the article, and approved the final draft.

## Data Availability

The raw measurements are available in the Supplemental File.

## Supplemental Information

Supplemental information for this article can be found online at http://dx.doi.org/10.7717/peerj.17655#supplemental-information.

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
