# Peer review of "The effects of meliponicultural use of Tetragonula laeviceps on other bee pollinators and pollination efficacy of lemon"

_PeerJ, doi:10.7717/peerj.17655_

## Round 0.1 · original submission · Major Revisions

Both reviewers suggested major revisions. Please, consider their comments and provide point-by-point response, when resubmitting new version of your manuscript.

**Language Note:** The review process has identified that the English language must be improved. PeerJ can provide language editing services - please contact us at [email protected] for pricing (be sure to provide your manuscript number and title). Alternatively, you should make your own arrangements to improve the language quality and provide details in your response letter. – PeerJ Staff

·

Basic reporting

The title of this manuscript needs a correction. Do you mean that T. laeviceps meliponiculture will affect the diversity, and foraging behavior of wild bee pollinators, and the pollination efficacy of Citrus limon?

You need to improve your English. You can check the correction with the yellow marks on the manuscript also you need to recheck the language of the whole manuscript.

You need The references are suitable for the manuscript, but you need to add more information about the reason why chose Citrus lemon. the complete comments were on the revised manuscript.

I think it has a similar pattern to your previous article on 2023 but on a different citrus. Some of the parameters are the same. The data and the procedure were missing, you need to improve the procedure.

Experimental design

The manuscript fulfills the aims and scope of PeerJ.

The research questions need to be improved.

the investigation was conducted based on previous research.

Something is missing from the method. you can check my comments on the manuscript, it must be clear.

Why was the observation on non meliponiculture only for one period, while meliponiculture was conducted for 2 periods?

How you can get the harvest number on non meliponiculture, while it only conducted for a month?

Validity of the findings

This research has a similarity with your previous research on another species of Citrus in different places. I think we were talking about Citrus, so the wild bee and the behavior can be similar.

The method and result of your previous article are clear. the PCA results of microclimate are relatively similar.

you did not describe the behavior of other foraging behavior clearly, especially for the pollination sequences (Figure 2).

the conclusion must be improved

Additional comments

This manuscript is good enough, but must improve your English and the procedure on methodology.
Please make sure that your manuscript is different from your previous article but you need to describe it clearly as the same as the article.

Good Luck.

Reviewer 2 ·

Basic reporting

The presentation of the manuscript is satisfactory. There is a need for English editing to enhance grammar and clarity. The authors have utilized enough recent and updated literature to support the presentation and discussion of the research.

The methodology is direct. Though, there is some information that needs to be added. The tables and figures are well-presented and relevant to the manuscript. It conforms to the acceptable format of the journal.

Experimental design

The introduction has clearly defined the importance or significance of the research. The manuscript adheres to acceptable scientific and ethical standards. The knowledge gaps have been identified as well as those that were not addressed by the research. These could have been presented as recommendations to entice readers to further venture into studies quite similar to the research.

Validity of the findings

The authors have presented their data well. However, language editing may assist in making their presentation clearer and easier to understand. Discussions and conclusions are well-stated, limited to the research, and commendable.

Additional comments

The research is novel and simple enough for young scientists to follow and replicate. There are just some inquiries that need to be clarified or addressed for better presentation and understanding.

Still, I commend the authors for engaging in these kinds of research and contributing information in the field of pollination ecology.

Annotated reviews are not available for download in order to protect the identity of reviewers who chose to remain anonymous.

---

## Round 0.2 · Minor Revisions

Please, consider few remaining amendments of one of the reviewer, check the annotated manuscript and provide further improvements.

·

Basic reporting

• The authors have improved their English
• The background has been improved, but they need to add some information
• The literature references were updated and they can support this research
• The methodology is well described, but there is some information needs to be added.
• The figures and tables are well-presented.

Experimental design

• The manuscripts fulfill the aims and scope of PeerJ
• This manuscript was conducted based on previous research, and the research questions are quite good.
• The knowledge gaps have been identified and the methods described in detail and informative to replicate

Validity of the findings

• The data of this manuscript are well-presented.
• Adding some information and language editing will improve this manuscript

Additional comments

• This research is simple to replicate by other scientists. Adding more information will improve your manuscript

---

## Round 0.3 · accepted · Accept

The authors did a very good job in providing minor revision or replied appropriately.